# Generating Action-conditioned Prompts for Open-vocabulary Video Action Recognition

Chengyou Jia*
School of Computer Science and
Technology, MOEKLINNS Lab,
Xi'an Jiaotong University
Xi'an, Shaanxi, China
cp3jia@stu.xjtu.edu.cn

Minnan Luo†
School of Computer Science and
Technology, MOEKLINNS Lab,
Xi'an Jiaotong University
Xi'an, Shaanxi, China
minnluo@xjtu.edu.cn

Xiaojun Chang‡
University of Science and Technology
of China
Hefei, Anhui, China
cxj273@gmail.com

Zhuohang Dang§
School of Computer Science and
Technology, Xi'an Jiaotong University
Xi'an, Shaanxi, China
dangzhuohang@stu.xjtu.edu.cn

Mingfei Han
ReLER Lab, AAII, University of
Technology Sydney
Sydney, New South Wales, Australia
hmf282@gmail.com

Mengmeng Wang¶
Zhejiang University of Technology,
College of Computer Science and
Technology, China
mengmewang@gmail.com

Guang Dai
SGIT AI Lab,
State Grid Corporation of China
Beijing, China
guang.gdai@gmail.com

Sizhe Dang
School of Computer Science and
Technology, Xi'an Jiaotong University
Xi'an, Shaanxi, China
darknight1118@stu.xjtu.edu.cn

Jingdong Wang
Baidu Inc
Beijing, China
wangjingdong@outlook.com

## Abstract

Exploring open-vocabulary video action recognition is a promising venture, which aims to recognize previously unseen actions within any arbitrary set of categories. Existing methods typically adapt pretrained image-text models to the video domain, capitalizing on their inherent strengths in generalization. A common thread among such methods is the augmentation of visual embeddings with temporal information to improve the recognition of seen actions. Yet, they compromise with standard less-informative action descriptions, thus faltering when confronted with novel actions. Drawing inspiration from human cognitive processes, we argue that augmenting text embeddings with human prior knowledge is pivotal for open-vocabulary video action recognition. To realize this, we innovatively blend video models with Large Language Models (LLMs) to devise Action-conditioned Prompts. Specifically, we propose the Action-Centric generation strategy to produce a set of descriptive sentences that contain distinctive features for identifying given actions. Building upon this foundation, we further introduce a multi-modal action knowledge alignment mechanism to align concepts in video and textual knowledge encapsulated within the prompts. Extensive experiments on various video benchmarks, including zero-shot, few-shot, and base-to-novel generalization settings, demonstrate that our method not only sets new SOTA performance but also possesses excellent interpretability.

## CCS Concepts

• **Computing methodologies → Activity recognition and understanding**.

## Keywords

Video Action Recognition, Open-vocabulary Learning, Large Language Models, Multi-modal Alignment

**ACM Reference Format:**
Chengyou Jia, Minnan Luo, Xiaojun Chang, Zhuohang Dang, Mingfei Han, Mengmeng Wang, Guang Dai, Sizhe Dang, and Jingdong Wang. 2024. Generating Action-conditioned Prompts for Open-vocabulary Video Action Recognition. In *Proceedings of the 32nd ACM International Conference on Multimedia (MM '24), October 28-November 1, 2024, Melbourne, VIC, Australia.* ACM, New York, NY, USA, 10 pages. https://doi.org/10.1145/3664647.3680690

*This work was completed during an internship at SGIT AI Lab.

†The corresponding author: Minnan Luo, School of Computer Science and Technology, Ministry of Education Key Laboratory of Intelligent Networks and Network Security, Xi'an Jiaotong University, Xi'an, 710049, China

‡Also with [1] Mohamed bin Zayed University of Artificial Intelligence (MBZUAI).

§Also with [2] Shaanxi Province Key Laboratory of Big Data Knowledge Engineering, Xi'an Jiaotong University, Xi'an, 710049, China.

¶Also with [3] SGIT AI Lab,State Grid Corporation of China.

## 1 Introduction

*Why can humans effortlessly recognize novel actions in videos, even with limited or no prior exposure to those specific actions?* For instance, one can easily identify the action of "making sushi" in Figure 1, despite having rarely witnessed the process of sushi preparation. This remarkable ability primarily arises from two factors. First, a comprehensive understanding of foundational actions allow

humans to quickly approximate a novel action, *e.g.*, familiarity with the general action of cooking aids in the recognition of making sushi as a culinary process. Secondly, auxiliary knowledge involved in specific actions serves as supplementary insights to enhance the recognition of novel unseen actions, *e.g.*, recognizing the chef's role, identifying raw fish slices, and capturing the distinctive rolling motion in "sushi making". Drawing inspiration from human cognitive behavior, we posit that an effective open-vocabulary video action model should mirror these two factors, thus empowering it to recognize any arbitrary actions without prior exposure.

Existing approaches [18, 27, 33, 46] always build open-vocabulary video action models on the foundation of pretrained Vision-Language (VL) models, such as CLIP [40]. The primary aim is to harness CLIP's robust generalization capabilities and extend them to the video domain [46]. This is achieved by calculating the similarity between query video and textual embeddings of various categories, with the highest similarity score indicating the matched category. The open-vocabulary capability stems from the fact that any action categories can be represented and matched through text. Building on this premise, a common thread among existing methods is the integration of temporal modeling to evolve the image encoder into a video encoder, with techniques like cross-frame interactions [18] or temporal attention [33]. These advancements [33, 34, 46] have been instrumental in improving temporal perception of seen actions.

Nevertheless, such approaches can be regarded as addressing only the first factor. While they excel in recognizing foundational actions, these methods generally fall short when tasked with recognizing novel actions [41]. In Figure 1, we illustrate these methods relying on basic, manually designed prompts such as "a video of making sushi" often yield lower CLIP match scores, lacking a clear mechanism to discern novel actions. Considering the second factor mentioned previously, it's crucial to equip the model with additional knowledge pertaining to novel actions. Such considerations should encompass multi-attribute descriptions like the scene of the action, involved elements, relevant props, and so forth. As shown in Figure 1, integrating this auxiliary knowledge into the text encoder enables it to anchor the visual content, thereby enhancing the model's ability to recognize novel actions. Motivated by these observations, our key insight is that ***distinct actions, especially novel actions that are previously unencountered, should be associated with their own set of knowledge-rich prompts***, which we term as ***Action-conditioned Prompts***. As shown in Figure 2, these prompts are distinct from more generic, hand-written prompts. They not only establish connections with foundational actions but also facilitate the recognition of novel actions through specialized knowledge.

However, manually crafting these prompts presents significant challenges: 1) the process is resource-intensive, both in time and cost, making it impractical for large sets of actions. 2) the variability in annotators' perceptions may result in inconsistent and subjective descriptions, inevitably introducing biases. To address these challenges, we innovatively blend video action models with Large Language Models (LLMs) to devise Action-conditioned Prompts. Specifically, we introduce a novel Action-Centric generation strategy, which begins by constructing *Hierarchical Attribute Graph* for video actions. This graph systematically defines pivotal attributes across the categories of Scene, Actor, and Body-related aspects, as depicted in Figure 3, which provides a solid foundation for the model

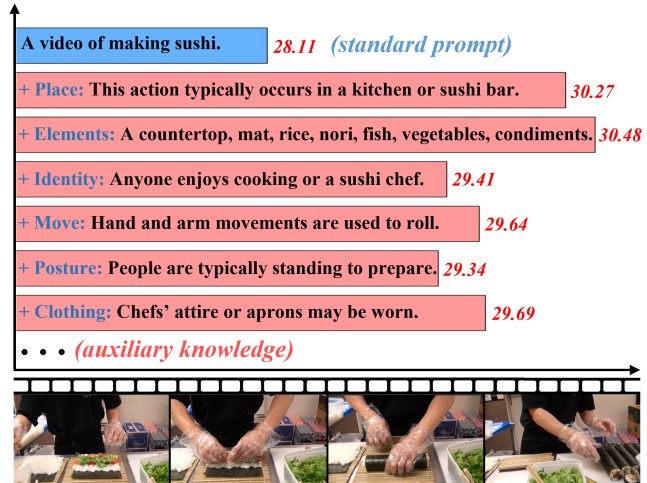

**Figure 1: The first line represents a traditional prompt, while the subsequent lines detail prompts that describe the action through multi-attributes. Scores on the right indicate CLIP match scores between the video and textual embeddings of corresponding prompts.**

to achieve a comprehensive understanding of video actions. The Action-Centric generation strategy then utilizes GPT-4 to generate knowledge-rich descriptions corresponding to each predefined attribute in the graph. These descriptions are then synthesized into multi-attribute Action-conditioned Prompts, processed through CLIP's text encoder and combined classifiers to advance open-vocabulary video action recognition. Building on these generating prompts, we further introduce a Multi-modal Action Knowledge Alignment (MAKA) mechanism to align visual concepts in video and textual knowledge within prompts. Specifically, we adopt the cross-modal late interaction, enabling the model to capture the fine-grained relevancy between each prompt and each frame.

Extensive experiments demonstrate that our method exhibits significant advancements over established baselines in various scenarios including zero-shot, few-shot, and base-to-novel generalization settings, as validated across five distinct video benchmarks. In all these extensive settings and metrics, our approach has consistently set new SOTA standards. Moreover, our method possesses excellent interpretability, providing a clear pathway to understanding how the model makes decisions when discerning actions through visual and textual cues. The contributions of this paper are as follows:

**1)** We argue that ***Action-conditioned Prompts*** with human prior knowledge is pivotal for open-vocabulary video action recognition. Accordingly, we propose a novel Action-Centric generation strategy to systematically devise these knowledge-rich prompts.

**2)** We further introduce a Multi-modal Action Knowledge Alignment (MAKA) to align visual concepts in video and textual knowledge within prompts. This mechanism provides fine-grained matching between videos and the corresponding prompts, enhancing the accuracy and interpretability of the action recognition.

**3)** Extensive experiments across a range of video benchmarks, including zero-shot, few-shot, and base-to-novel generalization settings, demonstrate that our method not only establishes new SOTA performance but also exhibits exceptional interpretability.

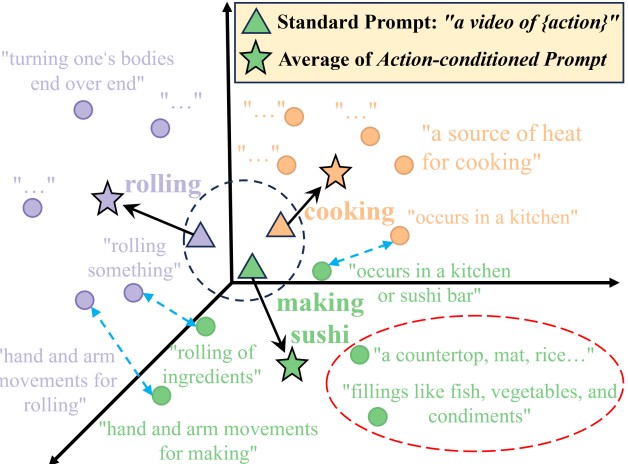

**Figure 2: Comparative visualization of text embeddings space for different prompts. The standard prompt yields limited information, concentrating embeddings within a confined area of the text space. In contrast, Action-conditioned prompts offer multi-attribute descriptions of actions, not only establishing connections with foundational actions but also providing the necessary knowledge to discern novel actions.**

## 2  Related Works

### 2.1  Video Action Recognition

The realm of video action recognition can be delineated into two principal methodologies: *uni-modal* and *multi-modal* approaches. Uni-modal methods are purely visual models, intensely focused on encoding both spatial and motion cues. Early approaches [10, 42, 45, 59] leveraged various low-level streams to capture temporal information, *e.g.*, optical flow and RGB differences. More recently, advanced mechanisms like 3D CNNs [8, 10, 13, 48] and video transformers [2, 29, 56] are proposed to model the long-range spatio-temporal relationships and have shown consistent improvements.

Complementing these uni-modal methods, the advent of Visual-Language (VL) pre-training [40] has catalyzed the emergence of multi-modal methods [60]. These innovative multi-modal approaches aim to harness CLIP's generalized VL representations for video recognition. A thread among such methods is the integration of temporal modeling to evolve the image encoder into a video encoder. For instance, Ni *et al.* [33] propose a cross-frame attention mechanism that explicitly exchanges information across frames. Pan *et al.* [34] develop 3D convolutional modules as adapters within the CLIP framework. Works such as ActionCLIP [46], STAN [28], ATM [52] also adopt similar strategies. Yet, these approaches generally underperform when tasked with identifying novel actions [41]. In contrast, our work introduces knowledge-rich action-conditioned prompts, aiming to enhance the recognition of novel actions.

### 2.2  Prompt Learning using LLM

The art of prompt engineering holds significant sway in refining the accuracy of language models [6, 14, 15, 44, 55] and vision-language models [40], which has incited extensive research into optimizing prompt formats. Early efforts ranged from assembling

manually crafted prompts [3] to devising learnable prompt tokens [23, 26, 61, 62]. Advancing beyond these, contemporary studies have leveraged prompts auto-generated by LLMs [31, 32, 37, 57], using them to create structured attribute lists that are reformulated into captions for use with CLIP. These methods demonstrate how the rich knowledge embedded in LLMs can effectively augment the perceptual capabilities of visual models. Considering that previous methods have primarily concentrated on fine-grained zero-shot image classification, there remains a lack of a systematic approach for exploring knowledge-rich prompts tailored to actions. This work seeks to complement the scarce literature by introducing innovative Action-conditioned Prompts for video action recognition.

## 3  Method

We first briefly overview the architecture of the CLIP model for video action recognition. Then, we elaborate on the critical component: Action-conditioned Prompts. Finally, we introduce the Multi-modal Action Knowledge Alignment mechanism.

### 3.1  Adapt CLIP for video action recognition

Given a video $V \in \mathbb{R}^{T \times H \times W \times 3}$ with $T$ frames and a text description $C$, where $V$ and $C$ are sampled from a set of videos and a collection of action category names respectively, we feed the $T$ frames into the video encoder $f_{\theta_v}$ and the text $C$ into the text encoder $f_{\theta_t}$ to obtain a video representation $v$ and a text representation $c$ correspondingly,

$$v = f_{\theta_v}(V), c = f_{\theta_t}(C). \tag{1}$$

The primary objective in fine-tuning clip for video action model lies in maximizing the similarity $sim(v, c)$ if $v$ and $c$ are correspondingly matched, and otherwise minimizing it. Typically, the similarity is calculated using cosine similarity,

$$sim(v, c) = \frac{\langle v, c \rangle}{\|v\| \|c\|}. \tag{2}$$

During inference, the similarity score is calculated between the given video and each action category, with the highest-scoring category being designated as the video's top-1 predicted classification.

### 3.2  Action-conditioned Prompts Generation

Figure 1 demonstrates the importance of supplementing models with expansive knowledge to augment action recognition. However, acquiring expert annotations is both cost-prohibitive and labor-intensive. It is also subject to individual biases, leading to inconsistent results. To address this, we harness the capabilities of Large Language Models (LLMs), *e.g.*, GPT-4, known for their extensive knowledge and versatility. Despite the absence of visual training data, LLMs can generate descriptions that capture action characteristics. This capability stems from the text data used in their training, which is authored by individuals imbued with visual knowledge, indirectly laying a foundation for visual recognition.

Although the application of LLMs to provide classificatory visual cues has previously been investigated [1, 31, 32], existing methods have predominantly concentrated on fine-grained image classification. For example, [31] focuses on the differentiation of various bird species, emphasizing static visual cues. Nonetheless, the analysis of video data demands a more holistic understanding of actions, incorporating not only visual elements but also the environmental

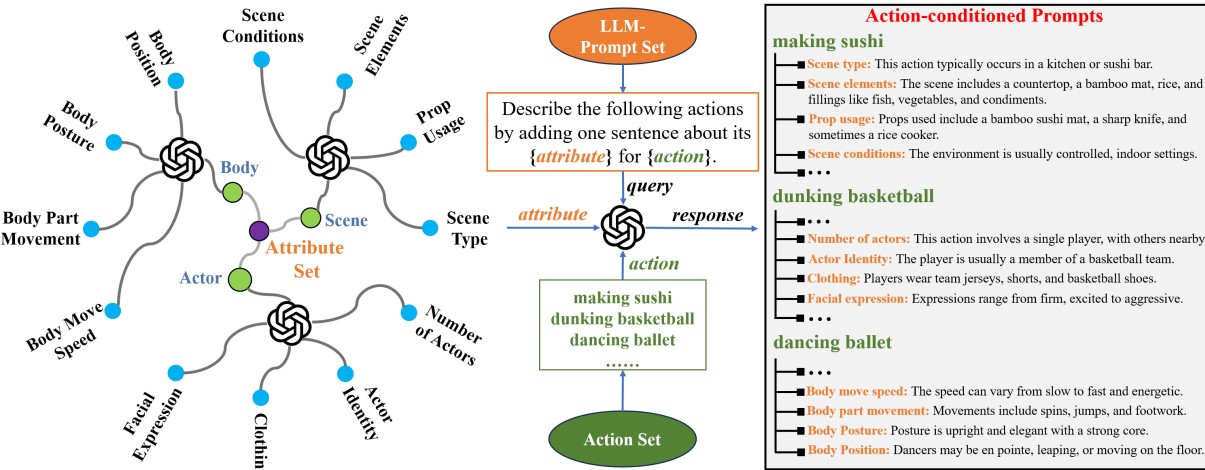

**Figure 3: Illustration of Action-conditioned Prompts generation workflow: On the left, the process of defining the *Hierarchical Attribute Graph* is visualized. The middle section depicts the querying process with LLMs, transforming action and attributes into structured prompts. On the right, we present sample snippets of the prompts generated.**

and temporal contexts related to actors and scenes. Therefore, we propose an ***Action-Centric generation strategy***. This strategy initially provides a comprehensive definition of video actions, fully capturing the distinctive features inherent to video actions. Building upon this foundation, by leveraging a series of defined core action attributes, we utilize LLMs to obtain specific multi-attribute action descriptions, culminating in the formation of Action-conditioned Prompts corresponding to the defined actions. We subsequently describe the specific steps of the Action-Centric generation strategy.

*3.2.1 **Hierarchical Attribute Graph of Video Actions**.* To discern critical attributes for action recognition and provide a complete definition of video actions, we initially categorize the components of an action into three fundamental aspects: Scene, Actor, and Body. These three dimensions collectively encapsulate all core contents of video actions, distinguishing them from the static attributes associated with still images. We then consult GPT-4 to ascertain which attributes within each aspect are necessary to differentiate actions. From GPT-4's responses, we select the four most representative attributes for each aspect. This process culminates in 12 core attributes, distributed across the three components, as displayed in the left part in Figure 3. Through this approach, we have developed a hierarchical video action attribute graph, which offers a comprehensive definition of video actions and lays a solid foundation for generating prompts. For a comprehensive understanding of the inquiries and GPT-4's responses, refer to the appendix.

*3.2.2 **Multi-attribute Action-conditioned Prompts**.* Building on the attributes defined in the hierarchical attribute graph, we further utilize GPT-4 to generate knowledge-rich descriptive sentences for each action category, forming the basis for our Action-conditioned Prompts. Specifically, we first construct a set of LLM-prompts. Then, for each LLM-prompt, we generate a suite of 12 distinct action-conditioned prompts, ensuring that every action is matched with tailored descriptive phrases, as depicted in Figure 3.

Taking the HMDB dataset as an example, which contains 51 categories, if 3 LLM-prompts are used for each category, the total

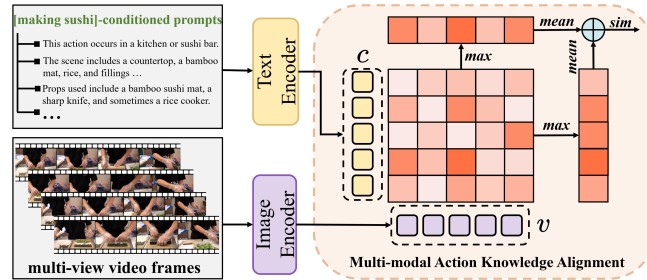

**Figure 4: Illustration of the MAKA mechanism.**

number of prompts generated would be $1886 = 3 \times 12 \times 51$. We constrain each prompt to a maximum of 30 tokens, truncating at the end of a sentence to ensure succinctness. Further refinements include the elimination of superfluous spaces and standardization of punctuation, enhancing consistency. Selected examples are showcased in the right segment of Figure 3. Before inputting each prompt into the action recognition model, we concatenate the action-conditioned prompt with a standard prompt format, "a video of {action}", to explicitly denote the represented action. For more LLM-prompts across all datasets and more detailed design specifics, please refer to the appendix in the supplementary material.

**Reliability of the generated prompts.** It is crucial to highlight that manual verification has been integrated into critical stages to safeguard the credibility of the generated content. The human selection of attributes was meticulously conducted through discussions among domain experts. To further ensure the reliability of the generated prompts, we manually conducted a rapid double-check of the corresponding prompts to eliminate any unreasonable prompts.

### 3.3 Multi-modal Action Knowledge Alignment

These generated action-conditioned prompts equip the model with a multifaceted understanding of actions. However, the challenge that follows is effectively aligning these prompts, which offer various perspectives, with corresponding visual concepts within the

videos. Previous methods temporally aggregate the embeddings of video frames, *e.g.*, mean pooling in [41] or attention pooling in [33], to yield a video-level representation. This approach, while prevalent, is not conducive to fine-grained alignment and leaves much to be desired in terms of nuanced representation and alignment strategies. To tackle this issue, we introduce a novel **Multi-modal Action Knowledge Alignment** mechanism to foster a more precise correspondence between text and video data.

To capture the multi-faceted features of videos, we implement a multi-view strategy [8, 13, 51] that samples multiple clips per video with several spatial crops, which allows the pre-trained vision-language model to encode each video into multiple frame embeddings, as illustrated in Figure 4. Further, inspired by [21, 58], we apply a cross-modal late interaction to model the fine-grained relevancy between each prompt and each frame.

Specifically, we define $n_v$ and $n_t$ as the count of frames for the video $V$ and the number of prompts for the category $C$, respectively. The visual features, denoted as $v = f_{\theta_v}(V) \in \mathbb{R}^{n_v \times d}$, and the prompt embeddings, denoted as $c = f_{\theta_t}(C) \in \mathbb{R}^{n_t \times d}$, are encoded accordingly. Here, $v$ and $c$ represent the normalized embeddings. Note that the current $C$ no longer refers to the category name but rather a series of action-conditioned prompts. The fine-grained similarity between $v$ and $c$ is computed via the following process:

For the $i$-th visual features in $v$, we assess its similarity across all prompt embeddings $c$, selecting the highest similarity by

$$\max_{0 \leq j < n_t} v_i^\top c_j, \tag{3}$$

which represents the maximum fine-grained similarity within the $C$ category. The video-to-category similarity is then the average of these maximum similarities across all visual features, given by

$$sim_{v2t}(v, c) = \frac{1}{n_v} \sum_{i=1}^{n_v} \max_{0 \leq j < n_t} v_i^\top c_j. \tag{4}$$

Conversely, for each $i$-th prompt embedding in $c$, we calculate its similarity with all visual features $v$, adopting the highest as the fine-grained maximum similarity. The category-to-video similarity is the average of these values,

$$sim_{t2v}(v, c) = \frac{1}{n_t} \sum_{i=1}^{n_t} \max_{0 \leq j < n_v} v_j^\top c_i. \tag{5}$$

We refine the cosine similarity computation (Equation (2)), in the training and inference processes by integrating both video-to-category and category-to-video similarities,

$$sim(v, c) = \frac{1}{2}(sim_{v2t}(v, c) + sim_{t2v}(v, c)). \tag{6}$$

For model fine-tuning, we utilize a standardized protocol as outlined in ViFi-CLIP [41]. More details of the training objectives and procedures are provided in the appendix.

Though inspired from [21, 58], it diverges significantly in its application of the cross-modal late interaction. Prior methods [58] have concentrated on fine-grained image-text matching at a token level, necessitating extensive training to relearn these associations. Such an approach does not contribute to enhancing generalizability. In contrast, the primary aim of our method is to establish frame-to-prompt correspondences, grounding similarity pairings in the natural image-text matching capabilities that are a forte of CLIP.

By leveraging this inherent strength, frame-to-prompt associations are more conducive to improved generalization.

## 4 Experiments

### 4.1 Experimental Setup

**Datasets.** We conduct experiments on five widely used video benchmarks: Kinetics-400 and 600 [7, 20], HMDB-51 [22], UCF-101 [43] and SSv2 [17]. See the appendix for more details of these datasets.
**Implementation details.** We use ViT-B/16 based CLIP [40] model for our experiments. Our adaptation of the CLIP model follows [41], with tailored modifications to the prompts and fine-grained similarity function used. We refer to our method as AP-CLIP (*Action-conditioned Prompt*). We use GPT-4 as the default prompt generator, and the default number of LLM-prompts is 3. Moreover, we align with previous methods [18, 33, 41, 46] for various settings including zero-shot, base-to-novel, few-shot, and fully-supervised. Specifically, we utilize 8 frames and employ multi-view inference incorporating 2 spatial crops and 2 temporal views. In the fully supervised setting, our approach extends to using 16 frames, combined with multi-view inference featuring 4 spatial crops and 3 temporal views, consistent with compared methods. More detailed prompts and training configurations are provided in the appendix.

### 4.2 Comparisons with State-of-the-art

*4.2.1 AP-CLIP Generalizes Well !* To demonstrate the open-vocabulary recognizing capabilities, we follow the benchmark in [41], evaluating models in two distinct settings: 1) **zero-shot setting** and 2) **base-to-novel setting**. The former primarily assesses the model's capacity to recognize novel actions across different datasets, while the latter tests its performance to recognize novel and rarer actions within the given dataset. Further details regarding these settings can be found in the appendix.
**(i) Zero-shot Setting**: We train the model on a large video action recognition dataset, Kinetics-400 and evaluate across different datasets, HMDB-51, UCF-101 and Kinetics-600. Results are presented in Table 1, where our model, AP-CLIP, is benchmarked against both uni-modal methods and other CLIP-based approaches. It's evident that even the vanilla CLIP demonstrates an impressive generalization performance as compared to uni-modal methods. Further analysis reveals that methods like ActionCLIP and XCLIP, which integrate additional temporal modules, may overfit on trained actions, thereby failing to show substantial generalization improvements. An alternative strategy, exemplified by ViFi-CLIP, involves merely fine-tuning the foundational CLIP model without incorporating external modules, yielding more promising generalization performance. Against this backdrop, our AP-CLIP also employs a straightforward fine-tuning of the CLIP model, incorporating *action-conditioned prompts* and the *multi-modal action knowledge alignment*. Table 1 demonstrates that AP-CLIP yields consistent performance improvements, with gains of +4.1%, +5.6%, and +2.2% on the HMDB-51, UCF-101, and Kinetics-600 datasets.

Furthermore, our method has demonstrated its adaptability. As a representative, we choose the current best competitor, Open-VCLIP [50], a robust model specifically designed for zero-shot action recognition. By integrating our action-conditioned prompts in place of its manual prompts, Open-VCLIP experienced a remarkable boost in

**Table 1: Zero-shot setting: We compare our AP-CLIP with uni-modal and CLIP-based approaches. Besides our AP-CLIP, generating action-conditioned prompts can be seamlessly integrated into the zero-shot approach [50]. Gains over previous methods are indicated in the bottom row. Methods marked with '*' are re-evaluated using their official code.**

| Method | HMDB-51 | UCF-101 | K600 |
|---|---|---|---|
| *Uni-modal zero-shot action recognition models* | | | |
| ASR [47] | 21.8 ± 0.9 | 24.4 ± 1.0 | - |
| ZSECOC [38] | 22.6 ± 1.2 | 15.1 ± 1.7 | - |
| UR [63] | 24.4 ± 1.6 | 17.5 ± 1.6 | - |
| E2E [5] | 32.7 | 48 | - |
| GCN [16] | - | - | 22.3 ± 0.6 |
| ER-ZSAR [9] | 35.3 ± 4.6 | 51.8 ± 2.9 | 42.1 ± 1.4 |
| *Adapting pre-trained VL models (ViT-B/16)* | | | |
| Vanilla CLIP [40] | 40.8 ± 0.3 | 63.2 ± 0.2 | 59.8 ± 0.3 |
| ActionCLIP [46] | 40.8 ± 5.4 | 58.3 ± 3.4 | 66.7 ± 1.1 |
| XCLIP [33] | 44.6 ± 5.2 | 72.0 ± 2.3 | 65.2 ± 0.4 |
| A5 [18] | 44.3 ± 2.2 | 69.3 ± 4.2 | 55.8 ±0.7 |
| VicTR [19] | 51.0 ± 1.3 | 72.4 ± 0.3 | - |
| ViFi-CLIP [41] | 51.3 ± 0.6 | 76.8 ± 0.7 | 71.2 ± 1.0 |
| AP-CLIP(ours) | **55.4 ± 0.8** | **82.4 ± 0.5** | **73.4 ± 1.0** |
| | +4.1 | +5.6 | +2.2 |
| Open-VCLIP [50] | 53.9 ± 1.2 | 83.4 ± 1.2 | 73.0 ± 0.8 |
| *+ Action prompts* | **57.0 ± 0.8** | **85.1 ± 1.2** | **74.4 ± 0.7** |
| | +3.1 | +1.7 | +1.4 |
| *Adapting pre-trained VL models (ViT-L/14)* | | | |
| BIKE* [54] | 50.2 ± 3.7 | 79.1 ± 3.5 | 68.5 ± 1.2 |
| Text4Vis [53] | 58.1 ± 5.7 | 85.8 ± 3.3 | 68.9 ± 1.0 |
| DiST [39] | 57.5 ± 1.6 | 74.9 ± 0.8 | - |
| Open-VCLIP [50] | 59.0 ± 0.6 | 87.6 ± 1.2 | 81.1 ± 0.8 |
| *+ Action prompts* | **60.0 ± 1.4** | **90.2 ± 0.4** | **81.9 ± 1.0** |
| | +1.0 | +2.6 | +0.8 |

its generalization capabilities, all without necessitating any retraining. Remarkably, under the ViT-L/14 CLIP model, this integration enhances Open-VCLIP to achieve groundbreaking performance, recording impressive scores of **60.0%**, **90.2%**, and **81.9%** across the three datasets, thereby establishing new SOTA in zero-shot action recognition. These results underscore significant generalization improvements from our action-conditioned prompts.

**(ii) Base-to-novel Generalization Setting**: In Table 2, we evaluate the generalization from base to novel classes on four datasets, K-400, HMDB-51, UCF-101 and SSv2. All methods were initially trained on well-established base classes, while the novel classes represented a realm of previously unencountered scenarios, *i.e.*, base and novel classes are disjoint. We adopted two distinct approaches for the latter three datasets: one leveraging the original CLIP parameters, and another utilizing parameters pre-trained on Kinetics-400. As shown in Table 2, AP-CLIP demonstrates noticeable gains in novel accuracy. Despite observing marginal reductions in base accuracy under certain conditions, our approach effectively balanced the trade-off between base and novel class performance, securing the highest

overall harmonic mean on all datasets. We also observed varied gains across different datasets. Temporally challenging datasets like SSv2 [17] showed limited improvements, whereas less temporally complex datasets like UCF [43] exhibited significant gains. In section 4.2.3, we provide a detailed discussion about the main reason for the smaller gains observed on the temporally challenging dataset. We also validate that our action-conditioned prompts can indeed facilitate larger improvements on such datasets.

*4.2.2  **AP-CLIP Specializes Well !*** Our investigation extends to AP-CLIP's efficacy in narrowing the domain gap within supervised video action recognition tasks. We evaluate its performance under two distinct data scenarios: 1) **few-shot setting**, where the number of training samples is limited [11, 12, 49], and 2) **fully-supervised setting**, where we have an abundance of samples. These settings help us to understand and evaluate the specialization performance of our approach under varying levels of data availability.

**(i) Few-shot Setting**: Table 3 delineates AP-CLIP's performance within a few-shot learning scenario, in comparison with other CLIP-based methodologies. AP-CLIP consistently exhibits performance improvements with increasing shots. Across both HMDB-51 and UCF-101 datasets, AP-CLIP surpasses all competing methods in each shot division (2, 4, 8, 16 shots). Notably, the advantage of our approach is more pronounced when training data is scant. The lesser training data provided, the more significant the improvement brought by our method. This suggests that our prompts provide more extensive knowledge, enabling the model to rapidly gain a deeper understanding of actions even with fewer examples.

**(ii) Fully-supervised Setting**: We compare the performance of AP-CLIP trained on Kinetics-400 with uni-modal video-specific models and other CLIP-based methods in Table 4. To ensure a fair comparison, results from methods employing ViT-L/14 have been excluded, with all CLIP-related models in this study based on the ViT-B/16 architecture. Despite AP-CLIP's primary intent for novel action recognition, Table 4 indicates its commendable applicability to fully-supervised tasks. Although it may not outstrip the more temporally intricate methods like UniFormerV2 [24] and DUALPATH [35], the margin of difference is not significant. Relative to baseline CLIP model fine-tuning [41], our approach delivers competitive performance. This substantiates our approach's utility in narrowing the domain gap between image and video modalities.

*4.2.3  **AP-CLIP Thrives under Temporally Challenging Scenarios!*** Our approach provides a comprehensive definition of video actions, emphasizing all essential aspects rather than merely the static attributes typically focused on in previous research [1, 31, 32]. Specifically, the Body-Related attributes defined in our method, such as Move Speed and Body Part Movement, are tailored to deepen the understanding of an action's movement pattern. However, results on temporally challenging datasets like SSv2 are somewhat disappointing, consistent with the performance of our baseline [41]. We identify that the principal limitation is the need for enhanced temporal perception within the visual encoder to more effectively synchronize with our prompts.

Therefore, to further validate the efficacy of our approach under Temporally Challenging Scenarios, we supplemented our method with baselines incorporating Temporal Modules (TM), as detailed in [50]. These modules extend the receptive field of the attention

**Table 2: Base-to-novel generalization: We compare the generalization ability of AP-CLIP with other models that adapt CLIP [40]. The values to the left of the "/" symbol indicate that models commence training from the native parameters of CLIP, while the right values denote that models are initially pre-trained on Kinetics-400, serving to bridge the modality gap. HM refers to harmonic mean which measures the trade-off between base and novel accuracy. Gains are shown in the bottom row.**

| Method | K-400 Base | Novel | HM | HMDB-51 Base | Novel | HM | UCF-101 Base | Novel | HM | SSv2 Base | Novel | HM |
|---|---|---|---|---|---|---|---|---|---|---|---|---|
| Vanilla CLIP [40] | 62.3 | 53.4 | 57.5 | 53.3 / - | 46.8 / - | 49.8 / - | 78.5 / - | 63.6 / - | 70.3 / - | 4.9 / - | 5.3 / - | 5.1 / - |
| ActionCLIP [46] | 61.0 | 46.2 | 52.6 | 69.1 / 69.0 | 37.3 / 57.2 | 48.5 / 62.6 | 90.1 / 85.6 | 58.1 / 75.3 | 70.7 / 80.1 | 13.3 / 8.1 | 10.1 / 8.7 | 11.5 / 8.4 |
| XCLIP [33] | 74.1 | 56.4 | 64.0 | 69.4 / 75.8 | 45.5 / 52.0 | 55.0 / 61.7 | 89.9 / 95.4 | 58.9 / 74.0 | 71.2 / 83.4 | 8.5 / 14.2 | 6.6 / 11.0 | 7.4 / 12.4 |
| A5 [18] | 69.7 | 37.6 | 48.8 | 46.2 / 70.4 | 16.0 / 51.7 | 23.8 / 59.6 | 90.5 / 95.8 | 40.4 / 71.0 | 55.8 / 81.6 | 8.3 / 12.9 | 5.3 / 5.7 | 6.4 / 7.9 |
| ViFi-CLIP [41] | 76.4 | 61.1 | 67.9 | 73.8 / **77.1** | 53.3 / 54.9 | 61.9 / 64.1 | 92.9 / **95.9** | 67.7 / 74.1 | 78.3 / 83.6 | 16.2 / 15.8 | 12.1 / 11.5 | 13.9 / 13.3 |
| AP-CLIP(ours) | **77.2** | **64.1** | **70.0** | **74.6** / 75.4 | **55.9** / **60.3** | **63.9** / **67.0** | **94.8** / 95.0 | **77.0** / **82.9** | **84.8** / **88.5** | **16.3** / **16.5** | **12.9** / 12.7 | **14.4** / **14.3** |
| | +0.8 | +3.0 | +2.1 | +0.8 / -1.7 | +2.6 / +3.1 | +2.0 / +2.9 | +1.7 / -0.9 | +9.3 / +7.6 | +6.5 / +4.9 | +0.1 / +0.7 | +0.7 / +1.2 | +0.5 / +1.0 |

**Table 3: Few-shot setting: The values on two sides of the "/" have the same meaning as in Table 2, one from CLIP's native parameters and the other pre-trained on Kinetics-400. Gains are indicated in the bottom row.**

| Model | HMDB-51 $K$=2 | $K$=4 | $K$=8 | $K$=16 | UCF-101 $K$=2 | $K$=4 | $K$=8 | $K$=16 |
|---|---|---|---|---|---|---|---|---|
| Vanilla CLIP [40] | 41.9 / - | 41.9 / - | 41.9 / - | 41.9 / - | 63.6 / - | 63.6 / - | 63.6 / - | 63.6 / - |
| ActionCLIP [46] | 47.5 / 54.3 | 57.9 / 56.2 | 57.3 / 59.3 | 59.1 / 66.1 | 70.6 / 76.7 | 71.5 / 80.4 | 73.0 / 87.6 | 91.4 / 91.8 |
| XCLIP [33] | 53.0 / 60.5 | 57.3 / **66.8** | 62.8 / 69.3 | 64.0 / 71.7 | 48.5 / 89.0 | 75.6 / 91.4 | 83.7 / 94.7 | 91.4 / 96.3 |
| A5 [18] | 39.7 / 46.7 | 50.7 / 50.4 | 56.0 / 61.3 | 62.4 / 65.8 | 71.4 / 76.3 | 79.9 / 84.4 | 85.7 / 90.7 | 89.9 / 93.0 |
| ViFi-CLIP [41] | 57.2 / 63.0 | 62.7 / 65.1 | 64.5 / 69.6 | 66.8 / 72.0 | 80.7 / 91.0 | 85.1 / 93.7 | 90.0 / 95.0 | 92.7 / 96.4 |
| AP-CLIP(ours) | **59.9** / **65.1** | **64.8** / **66.8** | **66.8** / **70.9** | **68.5** / **72.5** | **84.9** / **92.9** | **89.1** / **95.0** | **91.7** / **95.8** | **94.1** / **96.9** |
| | +2.7 / +2.1 | +2.1 / +0.0 | +2.3 / +1.3 | +1.7 / +0.5 | +4.2 / +1.9 | +4.0 / +1.3 | +1.7 / +0.8 | +1.4 / +0.5 |

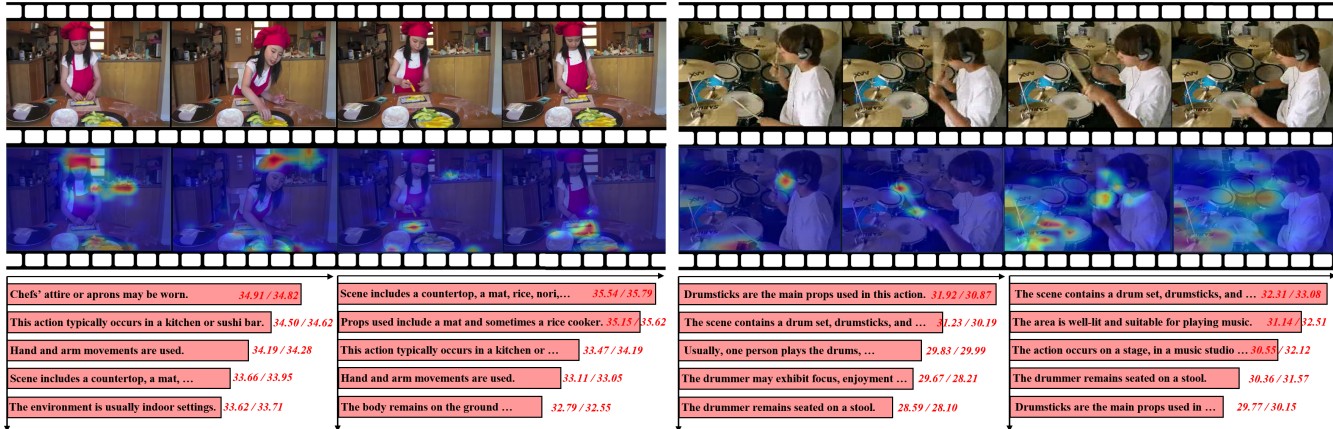

**Figure 5: The second row illustrates the visualization of visual attention maps from corresponding video frames. The third row displays the top-5 prompts for each video frame. For visual simplicity, two video frames are grouped together as they share the same top 5 prompts. The corresponding CLIP match scores are shown to the right of the bar graphs.**

mechanism to bolster temporal modeling capabilities. As demonstrated in Table 5, integrating temporal perception into our model leads to substantial improvements on the SSv2 dataset. Importantly, these benefits primarily arise from the Body attributes, which significantly aid in comprehending the action's movement pattern. This evidence highlights their critical role in temporally demanding contexts. Given that temporal modules introduce computational costs and provide limited benefits in non-temporal challenge scenarios, they were not included as a standard method of this paper.

## 4.3 Ablation Experiments

Extensive ablation experiments are conducted to demonstrate the efficacy of our AP-CLIP's components. We start from ViFi-CLIP [41] as the baseline, employing ViT-B/16 as its backbone. The model

**Table 4: Fully-supervised setting.**

| Method | Frames | Top-1 | Top-5 | Views |
|---|---|---|---|---|
| Uni-modal action recognition models | | | | |
| Uniformer-B [25] | 32 | 83.0 | 95.4 | 4 × 3 |
| TimeSformer-L [4] | 96 | 80.7 | 94.7 | 1 × 3 |
| Mformer-HR [36] | 16 | 81.1 | 95.2 | 10 × 3 |
| Swin-L [30] | 32 | 83.1 | 95.9 | 4 × 3 |
| ViViT-H [2] | 16 | 84.8 | 95.8 | 4 × 3 |
| UniFormerV2-B [24] | 8 | **85.6** | 97.0 | 4 × 3 |
| DUALPATH-B [35] | 32 | 85.4 | **97.1** | 1× 3 |
| Adapting pre-trained image VL models | | | | |
| ActionCLIP [46] | 32 | 83.8 | 96.2 | 10 × 3 |
| X-CLIP [33] | 16 | 84.7 | 96.8 | 4 × 3 |
| A6 [18] | 16 | 76.9 | 93.5 | - |
| STAN [28] | 16 | 84.9 | **96.8** | 1 × 3 |
| ViFi-CLIP [41] | 16 | 83.9 | 96.3 | 4 × 3 |
| AP-CLIP(ours) | 16 | **85.1** | 96.8 | 4 × 3 |

**Table 5: Results on temporally challenging SSv2 datasets.**

| Method | Base | Novel | HM |
|---|---|---|---|
| Base(ViFi-CLIP) | 16.2 | 12.1 | 13.9 |
| Base+TM | 16.7 (+0.5) | 12.4 (+0.3) | 14.2 (+0.3) |
| Ours | 16.3 (+0.1) | 12.9 (+0.8) | 14.4 (+0.5) |
| Ours+TM | **17.7** (+1.5) | **14.1** (+2.0) | **15.7** (+1.8) |
| Ours+TM w/o Body | 16.8 (+0.6) | 13.0 (+0.9) | 14.6 (+0.7) |

**Table 6: Analysis on different prompting strategies.**

| Method | HMDB-51 | UCF-101 | K-600 |
|---|---|---|---|
| Single [41] | 51.3 | 76.8 | 71.2 |
| Set [50] | 52.9 | 79.1 | 71.4 |
| Customized [37] | 53.1 | 78.8 | 71.8 |
| LLMs-Direct | 52.7 | 77.9 | 71.1 |
| AP(ours) | **54.2 (+1.5)** | **80.7 (+2.8)** | **72.4 (+1.3)** |

**Table 7: Ablation of the MAKA mechanism.**

| Method | HMDB-51 | UCF-101 | K-600 |
|---|---|---|---|
| Customized [37] | 53.1 | 78.8 | 71.8 |
| + MAKA | 53.9 (+0.8) | 79.9 (+1.1) | 72.2 (+0.4) |
| AP(ours) | 54.2 | 80.7 | 72.4 |
| + MAKA | **55.4 (+1.2)** | **82.4 (+1.7)** | **73.4 (+1.0)** |

is pre-trained on the Kinetics-400 dataset. Evaluation is carried out across various datasets, including HMDB-51, UCF-101, and Kinetics-600, to assess the impact of *action-conditioned prompts* and the *multi-modal action knowledge alignment* within our framework.

**(i) Is the "*action-conditioned prompts*" important?**

Table 6 assesses the effectiveness of different prompting strategies on model performance. These include the single prompt from ViFi-CLIP [41], a collection of manually crafted prompts in Open-VCLIP [50], and several approaches involving prompts generated by GPT. The generative strategies consist of Customized prompts in [37], action prompts directly generated by LLMs, and our specifically designed prompts. The findings suggest that LLM-generated prompts, with their rich knowledge base, not only reduce manual labor but also substantially bolster performance. Moreover, while directly utilizing the extensive knowledge from LLMs is a trivial solution, the Action-Centric generation strategy inherent in our action-conditioned prompt generation can better manage knowledge from various perspectives, resulting in better generalizability. This approach better mirrors human cognitive processes, crafting prompts that more effectively aid in action recognition.

**(ii) Is the "*multi-modal alignment*" important?**

The multi-modal alignment mechanism aims to align multi-attribute prompts with videos and is thus applicable exclusively to methods that utilize multiple prompts. Table 7 showcases that the incorporation of MAKA with both Customized prompts and our Action-conditioned Prompts results in uniform performance gains. This suggests that the multi-modal alignment facilitates the model's

development of a more comprehensive and nuanced comprehension of actions, thereby enhancing recognition performance.

## 4.4 Interpretability

Owing to the multi-attribute action-conditioned prompts and the multi-modal action knowledge alignment, our approach exhibits notable interpretability, which aids in elucidating the rationale behind the model's judgments. In Figure 5, we present the attention map visualizations of video frames alongside the score distributions for various attribute prompts. These visualizations reveal that different frames within a video garner varying focal points, which correspond to distinct prompts. For instance, as depicted in Figure 5 (left), initial frames focus primarily on the actor's clothing and the surrounding environment, where actor-related and scene-related prompts provide more clues for judgment. Conversely, subsequent frames shift attention to scene elements and props. Similarly, the example on the right also demonstrates a shift in focus, from props to the environment, and illustrates how well these aspects match with the corresponding prompts. These observations affirm that a model's access to diverse knowledge enhances its action recognition abilities. Crucially, they also highlight the importance of concept alignment within the video to corresponding prompts, a vital factor for interpretability. In the appendix, we provide a more detailed visualization of the frame-to-prompt correspondence.

## 5 Conclusion

In this work, we blend video models with Large Language Models (LLMs) to enhance open-vocabulary action recognition. Our strategy centers on generating Action-conditioned Prompts that enrich the textual embeddings in the CLIP model with human prior knowledge. Building on these knowledge-based prompts, we introduce a multi-modal action knowledge alignment mechanism to align concepts in video and knowledge encapsulated within the prompts. Extensive experiments not only demonstrate the effectiveness of our approach but also highlight its superior interpretability. By highlighting the significance of knowledge-based prompting, we expect that this research will catalyze additional investigation and innovation within the domain of action recognition.

## Acknowledgments

This work is supported by the National Nature Science Foundation of China (No. 62192781, No. 62272374), the Natural Science Foundation of Shaanxi Province (No. 2024JC-JCQN-62), the National Nature Science Foundation of China (No. 62250009, No. 62137002), Project of China Knowledge Center for Engineering Science and Technology, Project of Chinese academy of engineering "The Online and Offline Mixed Educational Service System for 'The Belt and Road' Training in MOOC China", and the K. C. Wong Education Foundation.

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
