# OpenReview forum: "Generating Action-conditioned Prompts for Open-vocabulary Video Action Recognition"
_acmmm.org/ACMMM/2024/Conference — MM2024 Poster_

### Official Review · Reviewer_T8B2 · 2024-05-21

**Rating:** 3
**Confidence:** 4

**Summary:**

This work proposes Action-conditioned Prompts for Open-vocabulary Video Action Recognition. By leveraging the rich knowledge of large language models (LLMs) to generate action descriptions, this method achieves good zero-shot recognition performance across several action recognition datasets.

**Strengths:**

1. The experiments are comprehensive, covering several commonly used action recognition datasets. Additionally, ablation studies and visual attention map visualizations are conducted.
2. This work is well-written with a clear logic, making it overall easy to understand.

**Limitations:**

1. The paper does not explain the differences from previous works that use LLMs to generate action descriptions for action recognition [1,2].
2. Has the impact of using different language models or generating varying numbers of descriptions on the experimental results been explored?
3. How did you address the issue of LLIM-generated descriptions potentially exceeding the 77-token limit imposed by CLIP's text encoder?
4. Have you considered that using LLIM to generate descriptions may introduce biases into the model due to the knowledge embedded in LLIM? For example, the model may associate cooking actions with kitchens, which could potentially affect the recognition of cooking actions in outdoor environments?

[1] Xiang W, Li C, Zhou Y, et al. Generative action description prompts for skeleton-based action recognition[C]//Proceedings of the IEEE/CVF International Conference on Computer Vision. 2023: 10276-10285.

[2] Lin W, Karlinsky L, Shvetsova N, et al. Match, expand and improve: Unsupervised finetuning for zero-shot action recognition with language knowledge[C]//Proceedings of the IEEE/CVF International Conference on Computer Vision. 2023: 2851-2862.

**Suitability:**

2

---

### Official Review · Reviewer_4uTS · 2024-05-23

**Rating:** 4
**Confidence:** 3

**Summary:**

This paper proposes an approach for open-vocabulary video action recognition. Here, they prompt the pre-trained LLM with the action-conditioned prompts and use the outputs to solve the open-vocabulary video action recognition task. They introduce the multi-modal action knowledge alignment mechanism to match the visual inputs with the LLM outputs. Experimental evaluation is performed on multiple standard benchmark datasets for action recognition, under zero-shot, few-shot, and base-to-novel generalization settings. Results show improvement over the current SOTA.

**Strengths:**

--> The paper is very well written. It is easy to read and understand.
--> The motivation and contributions of the paper are clearly explained.
--> Experimental evaluation is thorough.
--> SOTA results on multiple benchmark datasets.
--> Figures are very helpful in understanding the method.

**Limitations:**

--> The number of tokens is set to just 30 tokens and the prompts are of the form "a video of {action} {LLM output}", while the CLIP text encoder can process the inputs up to 77 tokens. Any reason for this?
--> Comparison between ViFi-CLIP and the proposed method w.r.t. to training procedure,  training time, and computation requirements would be helpful to gauge the trade-off between improvement in scores vs computation.
--> Zero-shot results on all the datasets, without fine-tuning CLIP on K400 using the LLM outputs can help understand the contribution of the LLM prompts.
--> Table 6 is not clear. Is this evaluation under a zero-shot setting?
--> The similarity scores shown in Figure 5 between the frames and LLM text are very similar. It is not obvious why the text "Scene includes a countertop, a mat, rice, ..." would have a lower similarity score to the initial frames and higher scores for the final frames. The same is the case with "Drumsticks are the main props used in...".  Drumsticks and drum sets are visible in all the frames.
--> While there is some novelty to the method, I think having no temporal modeling in the method proposed to solve video action recognition is a limiting factor.

**Suitability:**

3

---

### Official Review · Reviewer_uzP3 · 2024-05-28

**Rating:** 4
**Confidence:** 3

**Summary:**

The paper presents an interesting way to blend LLMs with video models to enhance open-vocabulary action recognition. To enrich the textual embedding of CLIP, the proposed approach generates action conditioned prompts with human prior knowledge. To this end, the authors introduce a multi-modal action knowledge alignment module which results in improved image-text alignment. Experimental results demonstrate the effectiveness of the proposed approach under various settings.

**Strengths:**

1. The paper proposes a novel action centric generation strategy to systematically devise knowledge rich prompts.
2. The authors introduce multi-modal action knowledge alignment to align the visual and the textual prompts. Empirical results show this approach helps enhancing the interpretability of action recognition.
3. The proposed approach outperforms existing approaches and sets new benchmark by achieving improved results in various tasks
4. The paper is generally well written and easy to follow.

**Limitations:**

1. The Hierarchical Attribute Graph of Video Actions step involves choosing 4 attributes for each aspect (scene, actor, body). Can the authors elaborate on how this threshold was decided by citing any empirical or theoretical arguments? How much does the proposed model's performance vary with varying number of attributes? Also how much latency does choosing more attributes incur in the existing set up?
2.  Do the authors perform any analysis on how sensitive is their approach on the correctness of GPT-4 employed attribute details extraction? Can the authors provide some analysis or some empirical support to their claims?
3. Can the authors provide some analysis on the additional semantic information obtained by such an approach and how much performance gain does this yield?
4. The overall set up looks like an ensemble of existing techniques. While leveraging GPT to squeeze additional information is well explored in the community, combining it
5. 'several spatial crops' (line 453) - can the authors elaborate more on how these spatial crops are determined?
6.  Although intuitive, I find lack of clarity in the description of MAKA and how it ensures fine-grained concept alignment? I might be missing this but do the authors claim this module injects both spatial as well as temporal fine-grained comprehension abilities into the system?
7. The experiment section lacks some important inspections: model's performance with different encoders (text and image),  sensitivity to GPT extracted information, fault tolerance etc.
8. The visualization on interpretability is particularly interesting. Can the authors add some discussions on the scope of some masking based cross-modal attention consistency enforcement modules? i.e., can an additional module be integrated into their set up that can better align the image-text features by means of providing strong spatial guidance?

**Suitability:**

3

---

### Official Review · Reviewer_6seA · 2024-06-27

**Rating:** 3
**Confidence:** 3

**Summary:**

The paper introduces a method for improving open-vocabulary video action recognition by integrating Large Language Models (LLMs) with video recognition systems to generate action-conditioned prompts. These prompts help the system recognize and interpret a broad range of actions, including those not seen during training, by embedding human prior knowledge into the model. This approach significantly enhances the model's ability to understand and categorize video actions, particularly in challenging scenarios like zero-shot and few-shot settings.

**Strengths:**

1.The paper is well-structured and clearly written, making it accessible to readers with varying levels of expertise in the field. The methodology is detailed thoroughly, allowing for reproducibility and deeper understanding of how the action-conditioned prompts are generated and utilized within the model.
2. The technical correctness of the approach is well-founded, utilizing a combination of LLMs with video recognition models to enhance the recognition capabilities through enriched textual embeddings.

**Limitations:**

1. The approach heavily relies on LLMs like GPT-4, which require substantial computational resources, making it less accessible for wider use.
2. Despite using LLMs, the method still involves significant manual verification to ensure the quality of the generated prompts, which is time-consuming and labor-intensive.
3. The overall complexity of the method may hinder its practical implementation and scalability.
4. The method shows limited improvements on temporally challenging datasets, such as SSv2, indicating insufficient handling of temporal dynamics. I'm not sure if I understand this correctly, but it seems that any model with text-image alignment could potentially utilize this method. Therefore, I hope the authors could experiment with additional backbone models with more related to temporal to broaden the applicability of the paper.

**Suitability:**

3

---

### Meta-Review · Area_Chair_exDd · 2024-06-30

**Recommendation:** Accept (Poster)
**Confidence:** 4

**Metareview:**

The paper tackles the challenging problem of open-vocabulary video action recognition. In a departure from previous approaches, priors about the actions are introduced by using LLM-generated descriptive sentences to enhance performance for the task. A multi-modal alignment mechanism is introduced to suitably align the text and visual modalities. The proposed approach performs well on various video benchmarks.

The broad sentiment for the paper is positive, although not by a big amount. Post-rebuttal, most reviewers retained their rating and one reviewer moved their review in the positive direction. The reviewers seemed concerned about the limited evaluation (bias arising from using a single LLM, image encoder) and lack of modelling the temporal aspect.	The paper provides an interesting approach for injecting action priors which has the potential to advance the capabilities for zero-shot video action recognition. Although reviewers raise valid concerns, the contributions in the paper are sufficient for establishing a strong baseline for future comparisons.

**I recommend acceptance of the paper**. The authors should address the concerns raised by the reviewers and improve the quality of final draft. The authors are encouraged to share their codebase for benefit of the community.